# Learning to Optimize via Dual space Preconditioning

## Abstract

Preconditioning an minimization algorithm improve its convergence and can lead to a minimizer in one iteration in some extreme cases. There is currently no analytical way for finding a suitable preconditioner. We present a general methodology for learning the preconditioner and show that it can lead to dramatic speed-ups over standard optimization techniques.

## 1 Introduction

Many problems arising in applied mathematics can be formulated as the minimization of a convex function $f : \mathbb{R}^d \to (-\infty, +\infty]$

$$\min_{x \in \mathbb{R}^d} f(x).$$

The resolution of an optimization problem is usually tackled using an optimization algorithm that produces a sequence of iterates converging to some minimizer of $f$ (Nesterov (2018)). The gradient descent algorithm is a standard optimization algorithm that converges linearly (*i.e* exponentially fast) if $f$ is regular enough. Preconditioned methods (Nemirovsky & Yudin (1983); Lu et al. (2016); Maddison et al. (2019)) are powerful optimization algorithms that converges linearly under weaker assumptions that gradient descent. The performance of these methods relies heavily on the preconditioning – the task of choosing the hyperparameter called the preconditioner. In the case of the Dual space Preconditioned Gradient Descent (DPGD) of Maddison et al. (2019), an optimal preconditioning can lead to convergence in one iteration.

The preconditioner is a function that has to be selected properly w.r.t. $f$ to obtain the desired linear convergence. Although Maddison et al. (2019) gives some hints to precondition DPGD, there is currently no analytical way for finding a suitable preconditioner.

In this paper, we propose to learn the preconditioner of DPGD using a neural network. We make the following contribution:

- We propose a supervised learning setting to learn the preconditioner of an optimization algorithm (DPGD, Maddison et al. (2019))
- We present a general methodology that allows to effectively learn the preconditioner while avoiding issues related to this task
- We implement this methodology in dimension one and $50$ and show that this can lead to dramatic speed-ups.

The remainder is organized as follows. The next section provides background knowledge on the DPGD algorithm. Then, in section 3 we present our supervised learning setting. Finally, we apply this methodology in section 4. Additional developments are provided in the appendix as well as postponed proofs.

## 2 Background on dual preconditioning

The main message of this section is that a good preconditioner is a preconditioner that is "close" to $\nabla f^\star$, where $f^\star$ is the Fenchel conjugate of $f$.

## 2.1 LEGENDRE FUNCTIONS

Before introducing DPGD algorithm, we need to define the class of functions $f$ that we aim to minimize. These functions are called Legendre convex functions and are studied in (Rockafellar, 1970, Section 26). We recall their definition and some of their properties. Given a function $f : \mathbb{R}^d \to (-\infty, \infty]$, the domain of $f$ is the set $\{x \in \mathbb{R}^d, f(x) < \infty\}$. We denote $D(f)$ the interior of the domain of $f$.

**Definition 1.** *The function $f$ is Legendre if $D(f)$ is not empty, and if*

- *the function $f$ is lower semicontinuous on $\mathbb{R}^d$, and is differentiable and strictly convex on $D(f)$,*

- *the gradient $\nabla f$ is coercive on $D(f)$: For every sequence $(x_i)_{i \in \mathbb{N}} \in D(f)$ converges to a point on the boundary of the domain of $f$, $\|\nabla f(x_i)\| \to \infty$.*

Given a Legendre function $f$, one can define its Fenchel conjugate $f^\star(y) = \sup_{x \in \mathbb{R}^d} \langle x, y \rangle - f(x)$.

**Proposition 1** (Rockafellar (1970)). *The function $f$ is Legendre if and only if its Fenchel conjugate $f^\star$ is Legendre, in which case $\nabla f$ is one-to-one between $D(f)$ and $D(f^\star)$. Moreover, $\nabla f$ is continuous in both directions, for every $x \in D(f)$, $\nabla f^\star(\nabla f(x)) = x$ and for every $y \in D(f^\star)$, $\nabla f(\nabla f^\star(y)) = y$.*

**Proposition 2** (Rockafellar (1970)). *If $f$ is Legendre, then it either has no minimizer, or one unique global minimizer $x_\star \in D(f)$. If $f$ admits a minimizer, then $\nabla f(x_\star) = 0$, $0 \in D(f^\star)$ and $x_\star = \nabla f^\star(0)$.*

## 2.2 DPGD ALGORITHM (MADDISON ET AL. (2019))

Consider a convex Legendre function $f : \mathbb{R}^d \to (-\infty, +\infty]$. To minimize $f$, the DPGD algorithm is written

$$x^+ = x - \gamma \left[ \nabla p(\nabla f(x)) - \nabla p(0) \right], \tag{1}$$

where $\gamma > 0$ and $p : \mathbb{R}^d \to (-\infty, +\infty]$ are parameters. DPGD is well defined over the set of Legendre convex functions $f$, see Maddison et al. (2019). The map $\nabla p$ is called a *non-linear preconditioner* of $\nabla f$. The preconditioning is the design of $\nabla p$.

## 2.3 CONVERGENCE THEORY OF DPGD

In the sequel, we consider a Legendre function $f$ admitting a minimizer $x_\star$. The preconditioning can dramatically affect the convergence of DPGD. Indeed, if $\nabla p = \nabla f^\star$, then DPGD algorithm converges in one iteration. To see this, note that in this case

$$x^+ = x - [x - \nabla p(0)] = \nabla p(0) = \nabla f^\star(0),$$

if $\gamma = 1$ and recall that $\nabla f^\star(0)$ is the minimizer of $f$ (Proposition 2). Of course, this preconditioning is unrealistic since computing $\nabla f^\star(0)$ is as hard as minimizing $f$.

It is more realistic to aim to find a preconditioner $p$ that satisfies the following conditions.

**Assumption 1** (Dual relative smoothness). *There exists $L \geq 0$ such that*

$$\langle \nabla p(x) - \nabla p(y), x - y \rangle \leq L \langle \nabla f^\star(x) - \nabla f^\star(y), x - y \rangle, \quad \forall x, y \in D(p). \tag{2}$$

**Assumption 2** (Dual relative strong convexity). *There exists $\mu > 0$ such that*

$$\langle \nabla p(x) - \nabla p(y), x - y \rangle \geq \mu \langle \nabla f^*(x) - \nabla f^*(y), x - y \rangle, \quad \forall x, y \in D(p). \tag{3}$$

If $p$ and $f^\star$ are twice differentiable, these assumptions can be written as

$$\mu \nabla^2 f^\star \overset{(3)}{\preceq} \nabla^2 p \overset{(2)}{\preceq} L \nabla^2 f^\star,$$

for the ordering $\preceq$ of nonnegative matrices. These assumptions mean that $\nabla p$ must be chosen close to $\nabla f^\star$ in some sense. Moreover, they are sufficient for the linear convergence of DPGD.

**Theorem 1** (Maddison et al. (2019), Informal). *If assumption 1 holds and if $p$ is Legendre, then the sequence of iterates of DPGD converges to $x_\star = \arg\min f$. If assumption 2 holds, then the convergence is linear.*

In comparison, gradient descent (and many other first order methods, Nesterov (2018)) converges for smooth convex functions $f$ (*i.e.* functions $f$ such that $\nabla f$ is Lipschitz continuous). Moreover, the convergence is linear if $f$ is strongly convex. Many functions satisfy assumption 1 (and assumption 2) for some preconditioner $p$, without being smooth (or strongly convex), see Maddison et al. (2019).

If $f$ is smooth and strongly convex, the convergence rate of gradient descent depends on the global condition number $\kappa$ of $f$ (defined as the Lipschitz constant of $\nabla f$ divided by the strong convexity parameter). If $\kappa$ is high, then the linear convergence of gradient descent is slow and $f$ is said ill-conditioned. The function $f$ is said well-conditioned else. When $f$ is ill-conditioned, it might be well-conditioned locally (*i.e.*, on small subsets of $\mathbb{R}^d$) but the rate of convergence of gradient descent only takes into account the global, high condition number. This is not the case for preconditioned algorithms ( Li & Malik (2016); Maddison et al. (2019)).

## 3 SUPERVISED LEARNING SETTING

In this section, we describe our methodology to learn $\nabla f^\star$.

### 3.1 FORMULATION

Our idea is to sample points $x_i \in D(f)$ according to some distribution and create the dataset $\{(\nabla f(x_i), x_i), i \in \{1, \ldots, n\}\}$ where the $\nabla f(x_i)$ represent the features and $x_i$ the labels. The feature space is $D(f^\star)$ and the label space is $D(f)$ (Proposition 1). Since $\nabla f^\star$ maps $\nabla f(x_i)$ to $x_i$ (Proposition 1), $\nabla f^\star$ can be seen as the solution of this supervised learning problem. We choose a neural net to modelize $\nabla f^\star$. To solve our supervised learning problem, we design a machine learning algorithm. The goal of the training of the machine learning algorithm is to minimize the theoretical risk $L_\mu(\theta)$

$$L_\mu(\theta) = \mathbb{E}_{X \sim \mu} \left( \ell(\mathrm{Model}_{\nabla f^*}(\nabla f(X), \theta), X) \right), \tag{4}$$

where $\mu$ is the distribution of the labels $X$, $\theta$ represents the parameters of the neural network, $\mathrm{Model}_{\nabla f^\star}(\nabla f(X), \theta)$ is the output of the neural network with input $\nabla f(X)$ and parameter $\theta$, and

$$\ell(\mathrm{Model}_{\nabla f^*}(\nabla f(X), \theta), X)$$

is the loss associated to one sample $(\nabla f(X), X)$.

To fully specify the machine learning algorithm, we need to fully define $L_\mu$ and the way we approximate it. We proceed by

- Choosing an architecture to specify the function $\mathrm{Model}_{\nabla f^*}$ (this is problem specific, see sections 3.2, 4.2)
- Giving a value for $\mu$ (section 3.3)
- Providing a way to approximate the expectation in (4) by a finite sum (section 3.4)
- Choosing a loss $\ell$ (this is problem specific, see section 4.2).

Once this is done, the algorithm is trained using Stochastic Gradient Descent (SGD) (Bottou et al. (2018)).

### 3.2 GENERAL SETUP

In this section we give a general description of the map $\mathrm{Model}_{\nabla f^*}$. According to recommendations in Bengio (2012), we first standardize the features $\nabla f(x_i)$ and the labels $x_i$. In other words, we apply a diffeomorphism (*i.e*, a one-to-one map which is continuously differentiable in both directions) $G : D(f^\star) \to [-0.5, 0.5]^d$ to the features $\nabla f(x_i)$ and a diffeomorphism $H^{-1} : D(f) \to [-0.5, 0.5]^d$ to the labels $x_i$. Then a neural network is used to learn a predictor from the standardized features to the standardized labels, see figure 1. One can obtain a label from a standardized label by applying the

inverse map $H$ of $H^{-1}$, which can be easily computed (in practice $H^{-1}$ is essentially an addition and a multiplication by a positive number), see section 4.

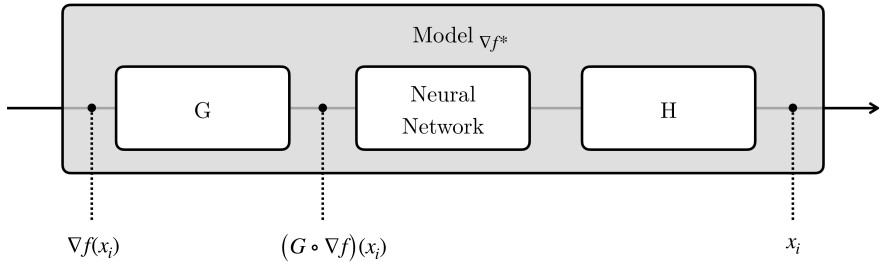

Figure 1: General model setup

### 3.3 CHOICE OF $\mu$

We found that the choice of the distribution $\mu$ that parametrized $L_\mu$ is critical for the performance of the machine learning algorithm. The reason is the following. Recall that we want to learn $\nabla f^\star$ to precondition DPGD algorithm whose goal is to minimize nonsmooth and non-strongly convex (or ill-conditioned) objectives $f$, see section 2. Let us imagine that we chose an uniform distribution for $\mu$. Then, the situation in dimension one is represented by the figure 2.

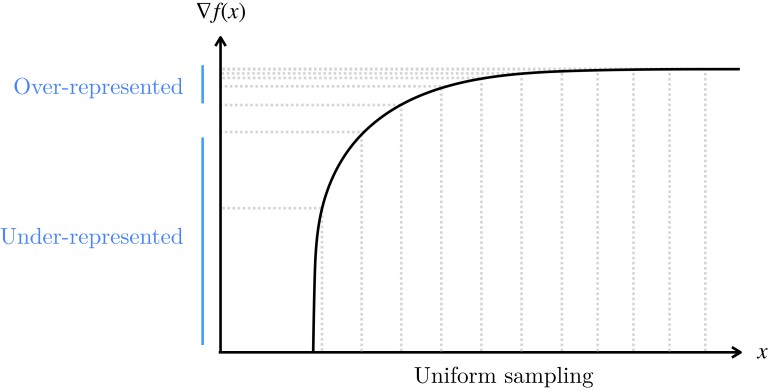

Figure 2: The curve of $\nabla f$ is typical of objectives $f$ that we want to minimize

The image distribution $\nu$ of $\mu$ by $\nabla f$ over-represents some areas of the feature space while other areas are under-represented. When the distribution of the features is as degenerated as $\nu$, it is obvious that machine learning algorithms cannot generalize well (see Bengio (2012); Mesnil et al. (2011)). Therefore, we have to choose $\mu$ carefully. The classical recommendation is to ensure that the input of the neural network (*i.e* the standardized features) are uniformly distributed (Mesnil et al. (2011)). In other words, we have to find $\mu$ such that, if $x_i \sim \mu$, then $G \circ \nabla f(x_i)$ is uniformly distributed over $[-0.5, 0.5]^d$.

This can be done by using the change of variable formula.

**Proposition 3.** *Assume that $\nabla f$ and $\nabla f^\star$ are differentiable. If $\mu$ is the distribution with density proportional to*

$$|\det(J_{G \circ \nabla f}(x))|$$

*with respect to Lebesgue measure (where $J$ denotes the Jacobian matrix), then the distribution of $G \circ \nabla f(X)$ where $X \sim \mu$ is uniform over $[-0.5, 0.5]^d$.*

In the sequel, $\mu$ is the distribution whose density is given by Proposition 3. Inspecting figure 2, this seems reasonable since we expect $X \sim \mu$ to have a high density when $\nabla f$'s slope is large and a low density when $\nabla f$'s slope is small.

**Remark 1.** *It is worth noting that the evaluation of $\mu$ doesn't require any inversion as in the inverse transform sampling technique. Besides, second order optimization algorithms usually require (an approximation of) the inverse of the Hessian matrix, which we don't need.*

### 3.4 ESTIMATION OF THE LOSS

At this step, $\mu$ is chosen and $L_\mu$ has to be minimized by SGD. As usual, $L_\mu$ cannot be computed in closed form. The classical idea is to approximate $L_\mu$ by an empirical mean involving samples from the distribution $\mu$. Sampling from $\mu$ turns out to be difficult in our case. Indeed, the use of rejection-based MCMC methods lead to a high rejection rate. Since we can evaluate the density of $\mu$, we suggest to use importance sampling instead.

**Proposition 4.** *If $x_i \sim \pi$ where the density of $\pi$ is positive, then*

$$\mathbb{E}\left(\frac{1}{n}\sum_{i=1}^{n}\ell(\mathrm{Model}_{\nabla f^*}(\nabla f(x_i), \theta), x_i)\frac{\mu(x_i)}{\pi(x_i)}\right) = L_\mu(\theta),$$

*where $\mu(x)$ (resp. $\pi(x)$) denotes the density of $\mu$ (resp. $\pi$) with respect to Lebesgue measure.*

We call the map $\theta \mapsto \frac{1}{n}\sum_{i=1}^{n}\ell(\mathrm{Model}_{\nabla f^*}(\nabla f(x_i), \theta), x_i)\frac{\mu(x_i)}{\pi(x_i)}$ the empirical risk. Proposition 4 states that if $x_i \sim \pi$, then the empirical risk is an unbiased estimator of the theoretical risk. Since this estimator will be minimized using SGD, we only need to know it up to a constant factor. Therefore we can get rid of the $1/n$ factor and of constant factors defining the densities of $\mu$ and $\pi$. The distribution $\pi$ is called the proposal distribution.

### 3.5 RELATED WORKS

Learning an optimization algorithm has already been proposed in the literature. In Li & Malik (2016); Andrychowicz et al. (2016) the authors propose to learn a better optimization algorithm by observing its execution over a feature space of objective functions. This outperforms existing hand-engineered algorithms in terms of convergence speed, but require to know the performance of some algorithms over some objectives. Using optimization literature knowledge, our approach allows to reduce the problem to a supervised learning setting in $\mathbb{R}^d$, that we can solve with a neural net. In its philosophy, our approach can also be related to deep learning methods for inverse problems used in signal processing, see *e.g* Lucas et al. (2018).

## 4 NUMERICAL EXPERIMENT

In this section, we run DPGD algorithm after learning $\nabla f^\star$. We first consider a one dimensional toy problem with a ground truth involving power functions as in Maddison et al. (2019). Then ill-conditioned logistic losses are considered. These functions cannot be minimized (efficiently) by gradient descent. We call LDPGD our approach (for Learned DPGD).

### 4.1 POWER FUNCTIONS AND LOGISTIC LOSS

Two class of objective functions are considered. Power functions are defined by

$$f : x \mapsto \frac{1}{a}\|Ax - c\|^a, \quad \text{where} \begin{cases} a \in (2, +\infty) \\ c \in \mathbb{R}^d \\ A \text{ is a matrix.} \end{cases} \tag{5}$$

These functions are nonsmooth and non-strongly convex, therefore gradient descent is not guaranteed to minimize them. Power function minimization is the main example of Maddison et al. (2019).

We also consider the logistic loss

$$f : x \mapsto \frac{1}{n}\sum_{i=1}^{n}\log\left(1 + \exp\left(-b_i\langle a_i, x\rangle\right)\right) + \frac{r}{2}\|x\|^2, \tag{6}$$

where $r > 0$ and for every $i$, $b_i \in \{-1, 1\}^n$ and $a_i \in \mathbb{R}^d$. These functions are strongly-convex and smooth. It is easy to control the condition number of $f$:

**Lemma 1.** *Consider the logistic loss $f$ and the matrix $A$ whose lines are the $a_i$, and denote $\lambda_{\max}$ the largest eigenvalue of the real symmetric matrix $A^T A$. Denote $L$ the smoothness parameter of $f$[1] and $\kappa$ the condition number of $f$. Then,*

$$L \leq L' = \frac{\lambda_{\max}}{4n} + r, \quad and \quad \kappa \leq \frac{\lambda_{\max}}{4nr} + 1.$$

## 4.2  EXPERIMENTAL SETUP

We use a three layers neural network, with leaky ReLU activations and respectively 256 and 128 neurons in the two hidden layers. The square loss is used $\ell(y, x) = \frac{1}{2}\|y - x\|^2$. The proposal distribution $\pi$ for the importance sampling is taken gaussian. In the case of power functions, the features are rescaled using the map

$$\text{log-rescaling} : x \mapsto \begin{cases} \log(1 + x) \text{ if } x > 0 \\ -\log(1 - x) \text{ else.} \end{cases}$$

Justification for this choice is provided in the appendix (section D). Then the rescaled features and the labels are normalized to have zero mean and unit variance. The general architecture is summarized in figure 3.

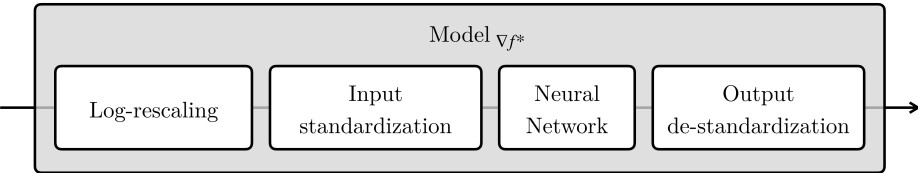

Figure 3: Model with log-rescaling

The two first blocks correspond to the map $G$ and the last block to the map $H$ of section 3.2.

The model is trained with a dataset of 1000 samples.

## 4.3  RESULTS FOR POWER FUNCTIONS

Consider $f$ a power function (5) where $d = 1$, $a = 50$, $c = 50$ and $A$ is randomly chosen. We train the model during 100 epochs.

The next figure represents the performance of the machine learning algorithm we developed to learn $\nabla f^*$ (note the scale of the inputs of $\nabla f^*$, see section D).

---

[1]*i.e*, the Lipschitz constant of $\nabla f$

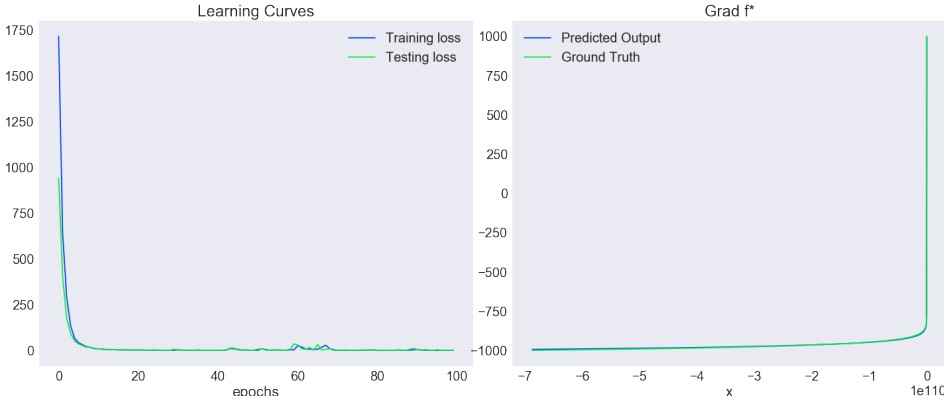

Figure 4: Learning $\nabla f^*$. Left: Learning curves. Right: predictions.

We see that $\nabla f^\star$ is accurately learned the machine learning algorithm. This approximation of $\nabla f^\star$ is then used as a preconditioner of DPGD. The value of the objective function and the value of the iterates while running LDPGD is plotted in the next figure. The stepsize is set to $1$ during $10$ steps and LDPGD is initialized at $x_0 = 100$.

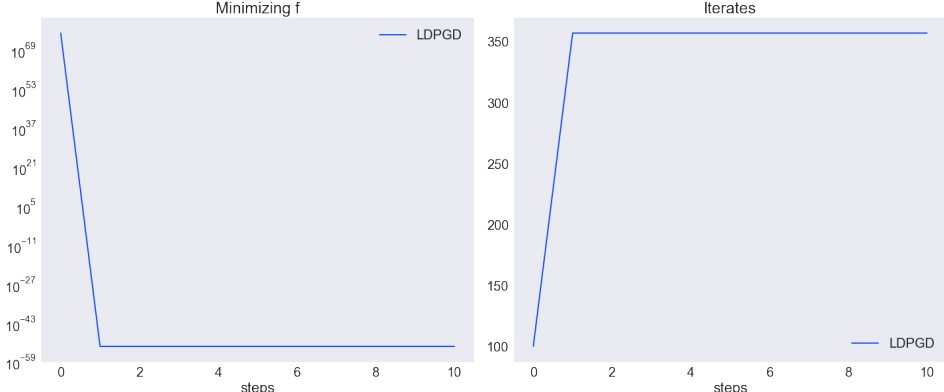

Figure 5: Minimizing a one dimensional power function. Left: Objective function values. Right: iterates values.

The iterates of gradient descent always diverge (no matter the stepsize choice). On the contrary, LDPGD iterates quickly converge to $357.1$ (with $\nabla f(357.1) = 3.8 \times 10^{-14}$) which is close to the actual minimizer ($356.4$ in this example).

### 4.4 RESULTS FOR LOGISTIC REGRESSION

Consider $f$ a logistic loss (6) with $d = 50$, $n = 1000$, and condition number $\kappa$. We train the model during $100$ epochs. and represent the evolution of the objective function and the iterates while running LDPGD (after learning $\nabla f^\star$) and gradient descent (GD).

The algorithms are initialized at $(50, 50, 50)^T$ and we use a stepsize for which GD converges linearly[2]. This is done for several logistic losses with worsening condition numbers. To vary $\kappa$, we vary the parameter $r$ that controls the strong convexity of the objective $f$ (see Lemma 1). Intuitively, if $r$ is high then $f$ is well-conditioned and if $r$ is low $f$ is ill-conditioned. The left figure represents the iterates (each curve corresponds to the evolution of one coordinate of the iterates) and the right figure represents the objective fonction values while running both algorithms.

---

[2]GD converges linearly if the stepsize is smaller than $1/L$, where $L$ is the Lipschitz constant of $\nabla f$. Therefore, it is enough to take a stepsize smaller than $1/L'$ where $L'$ is defined in Lemma 1

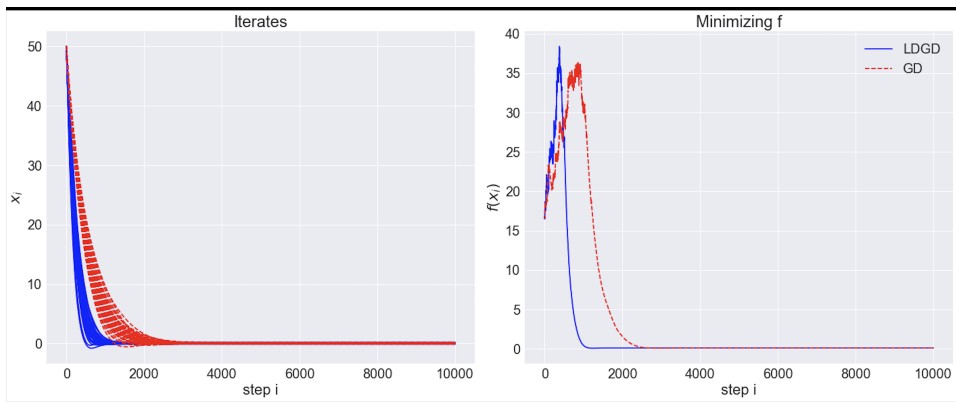

Figure 6: Logistic regression with $r = 0.5$

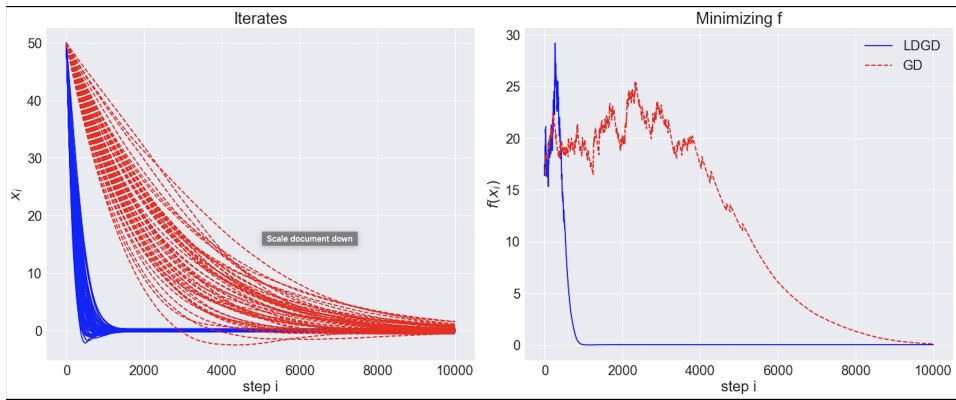

Figure 7: Logistic regression with $r = 0.1$

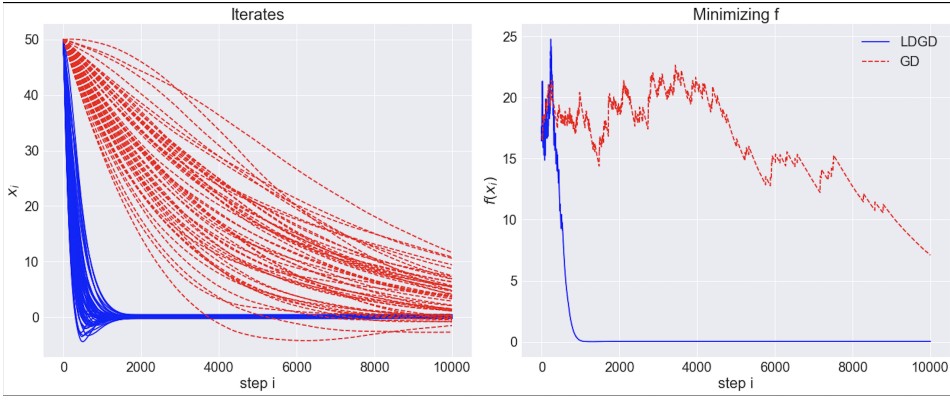

Figure 8: Logistic regression with $r = 0.05$

Our careful preconditioning allows LDPGD to outperform GD especially when the objective is ill-conditioned, as predicted by the theory, see section 2.

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

# Appendix

## CONTENTS

## A    PROOF OF PROPOSITION 3

Recall that $G$ is a diffeomorphism and that $f$ is Legendre. Let $\mathcal{U}$ the uniform distribution over $[-0.5, 0.5]^d$ and denote $q$ its density with respect to Lebesgue measure. Under the assumption of Proposition 3, $\nabla f : D(f) \to D(f^\star)$ is a diffeomorphism. Therefore, the map $M = G \circ \nabla f : D(f) \to [-0.5, 0.5]^d$ is also a diffeomorphism and its inverse map is $M^{-1} = \nabla f^\star \circ G^{-1} : [-0.5, 0.5]^d \to D(f)$. Our problem is to find a distribution $\mu$ supported by $D(f)$ such that $M(X) = G \circ \nabla f(X) \sim \mathcal{U}$ if $X \sim \mu$. Such $\mu$ satisfies for every nonnegative measurable function $\phi$,

$$
\begin{aligned}
\int \phi(x) d\mu(x) &= \int \phi(M^{-1} \circ M(x)) d\mu(x) \\
&= \mathbb{E}_{X \sim \mu}(\phi(M^{-1} \circ M(X))) \\
&= \mathbb{E}_{U \sim \mathcal{U}}(\phi(M^{-1}(U))) \\
&= \int \phi(M^{-1}(u)) q(u) du \\
&= \int_{D(f)} \phi(M^{-1}(M(x))) q(M(x)) |\det(J_M(x))| dx \\
&= \int_{D(f)} \phi(x) q(M(x)) |\det(J_M(x))| dx,
\end{aligned}
$$

where $J_M$ denotes the Jacobian matrix of $M$ and the penultimate equality follows from the change of variable formula (which is allowed because $M$ is a diffeomorphism). Therefore, the only possible solution to our problem is the distribution $\mu$ with density proportional to $|\det(J_M(x))|$ if $M(x) \in [-0.5, 0.5]^d$ (which is always satisfied by definition of $M$) and $0$ else. One can check that this density solves the problem.

## B    PROOF OF PROPOSITION 4

We apply the importance sampling principle to the measurable nonnegative function (the parameter $\theta$ is fixed)
$$
\phi(x) = \ell(\text{Model}_{\nabla f^*}(\nabla f(x), \theta), x).
$$
We have
$$
\mathbb{E}_{Y \sim \mu}(\phi(Y)) = \mathbb{E}_{X \sim \pi}\left(\phi(X) \frac{\mu(X)}{\pi(X)}\right) = \mathbb{E}\left(\frac{1}{n} \sum_{i=1}^n \phi(x_i) \frac{\mu(x_i)}{\pi(x_i)}\right),
$$
where $x_i \sim \pi$.

## C    PROOF OF LEMMA 1

Denote $L$ the smoothness constant of $f$ (*i.e* the Lipschitz constant of its gradient) and $\lambda$ its strong convexity parameter. For every $x \in \mathbb{R}^d$,

$$
\begin{aligned}
\nabla f(x) &= \frac{1}{n} \sum_{i=1}^n -b_i a_i \frac{1}{1 + \exp(b_i \langle a_i, x \rangle)} + rx \\
\nabla^2 f(x) &= \frac{1}{n} \sum_{i=1}^n b_i^2 a_i a_i^T \frac{\exp(b_i \langle a_i, x \rangle)}{(1 + \exp(b_i \langle a_i, x \rangle))^2} + rI_d,
\end{aligned}
$$

where $I_d$ is the identity matrix. Note that for all $a \neq 1$, $\frac{a}{(1+a)^2} \leq \frac{1}{4}$, and that $b_i^2 \leq 1$. It follows that

$$
\nabla^2 f \preceq \frac{A^T A}{4n} + rI_d.
$$

Therefore, $L \leq \frac{\lambda_{\max}}{4n} + r$. Since $\kappa = L/\lambda$, the result follows from the fact that $r \leq \lambda$ (see Equation 6).

# D    LOG-RESCALING

For power functions, $\nabla f(x)$ can range in $\left[-10^{100}, 10^{100}\right]^d$ if $x$ is sampled from a gaussian. Assume $d = 1$ that we use *Doubles* precision in the implementation. This means that all the values we manipulate are stored with a precision of approximately 15 decimal places.

If we naively standardize $(\nabla f(x_i))_{i \in [1,n]}$ to have zero mean and unit variance, we rescale numbers in the range $[-10^{100}, 10^{100}]^d$, to the range $[-0.5, 0.5]^d$. This means that a precision of $10^{-15}$ in the standardized scale represents a precision of $10^{85}$ in the original scale.

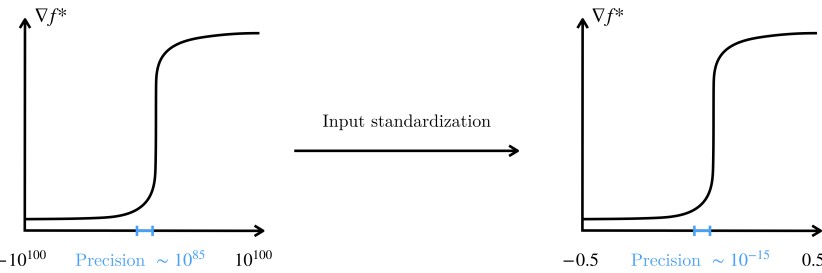

In practice, this loss of precision means that inputs that are different before standardization, might be indistinguishable after the standardization. As a result, the neural network might receive many different outputs for a single input value.

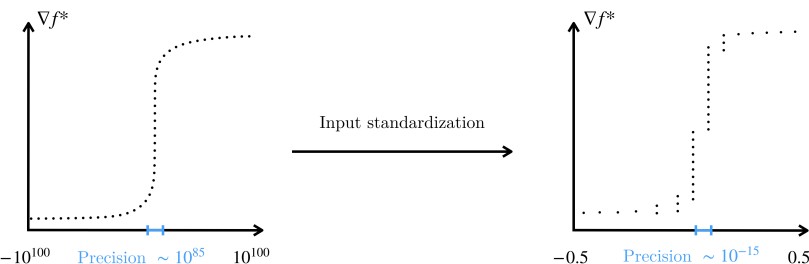

As it tries to minimize the empirical risk, the model will tend to associate to input values an average of the corresponding outputs. As a consequence, the regressed model will be a piecewise constant function, and will be constant on subsets of diameter $10^{85}$.

This is especially problematic for inputs close to 0. Let us denote $\mathcal{N}_0$ the neighborhood of 0 on which the regressed model is constant equal to $\text{Model}_{\nabla f^*}(0, \theta)$. Any $x \in D(f)$ for which $\nabla f(x) \in \mathcal{N}_0$ is a *fixed point* of the DPGD alogorithm:

$$
\begin{aligned}
x^+ &= x - \frac{1}{L_p} \left[ \text{Model}_{\nabla f^*}(\nabla f(x), \theta) - \text{Model}_{\nabla f^*}(0, \theta) \right] \\
&= x - \frac{1}{L_p} [0] \\
&= x
\end{aligned}
$$

The consequence is very undesirable: the solution offered by the DPGD alogorithm might be $x$ even though $\nabla f(x) = \mathcal{O}(10^{85})$ is far from zero.

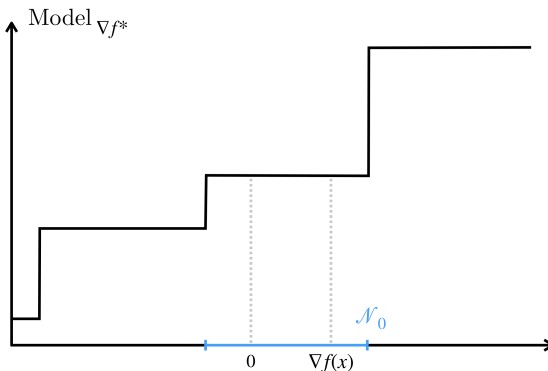

Figure 9: Regressed model on the imprecise dataset

As mentioned above, the lack of precision due to a naive standardization of the features is especially problematic around $0$. We therefore propose to preserve the scale around small values more than around large values. This can be achieved by applying the following transformation to the features before normalizing them (to have zero mean and unit variance):

$$\text{log-rescaling} : x \mapsto \begin{cases} \log(1 + x) \text{ if } x > 0 \\ -\log(1 - x) \text{ else.} \end{cases}$$

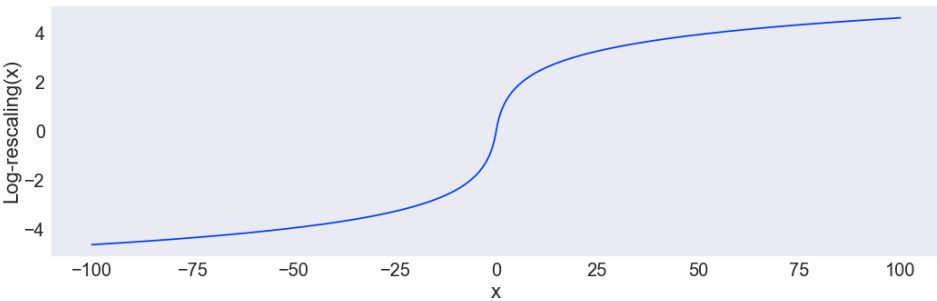

Figure 10: Log-rescaling.

