# OpenReview forum: "Learning to Optimize via Dual space Preconditioning"
_ICLR.cc/2020/Conference — Reject_

### Official Review · AnonReviewer2 · 2019-10-23
**Official Blind Review #2**

**Rating:** 3

**Review:**

[Update after rebuttal period]
I have read the response,  my confusion in the original reviews cannot be answered satisfactorily. Therefore, I keep my initial scores.


[Original reviews]
Firstly, the motivation of the proposed method is not convincing for me. The authors want to propose a general methodology for learning precondition by supervised learning setting. However the method in practice, the x is a complex distribution, it is difficult to handle the map between the gradient and the x. This method proposes log-scaling, but it needs to be stored with a precision of approximately 15 decimal places and the regressed model will be a piecewise constant function, which is very computationally time-consuming.

Secondly, the experimental results are not sufficient for evaluation. This paper shows two
The experimental result which includes the result of power function and the logistic function. But
It is not clear that the whole process of dual space preconditioned method with the model of computation of precondition given. And without quantitative results given, it is not convincing the “dramatic” speedups” of these methods, because the surprising training process is off-line and time-consuming. On the other hand, because of the different forms of the convex objective function, the network will train for the specific convex objective functions. In my opinion, it is not a general method to lead to dramatically speed up.

Finally，the function of x and the gradient is complex, it is difficult to predict the relationship by using a simple network,


**Experience Assessment:**

I have published in this field for several years.

**Review Assessment: Checking Correctness Of Derivations And Theory:**

I assessed the sensibility of the derivations and theory.

**Review Assessment: Checking Correctness Of Experiments:**

I assessed the sensibility of the experiments.

**Review Assessment: Thoroughness In Paper Reading:**

I read the paper at least twice and used my best judgement in assessing the paper.

---

> ### Author Response · Authors · 2019-11-14
> **Response to Review #2**
>
> Thank you for your careful analysis and feedback. We are adding more experiments and benchmarks in response to your comments.
>
>
> Q: Firstly, the motivation of the proposed method is not convincing for me. The authors want to propose a general methodology for learning precondition by supervised learning setting. However the method in practice, the x is a complex distribution, it is difficult to handle the map between the gradient and the x. This method proposes log-scaling, but it needs to be stored with a precision of approximately 15 decimal places and the regressed model will be a piecewise constant function, which is very computationally time-consuming.
> A: We’re not sure we understand this comment. Our motivation is to learn a pre-conditioner for the Dual space Pre-conditioning of the Gradient Descent method: there is currently no existing method to find such a pre-conditioner. You mention the fact that the regressed model might be piecewise constant. The log-rescaling precisely addresses this issue.
>
>
> Q: Secondly, the experimental results are not sufficient for evaluation. This paper shows two
> The experimental result which includes the result of power function and the logistic function. But
> It is not clear that the whole process of dual space preconditioned method with the model of computation of precondition given. And without quantitative results given, it is not convincing the “dramatic” speedups” of these methods, because the surprising training process is off-line and time-consuming. On the other hand, because of the different forms of the convex objective function, the network will train for the specific convex objective functions. In my opinion, it is not a general method to lead to dramatically speed up.
> A: In some cases, there might not be an alternative to the proposed method. This justifies using this method even though it might be time-consuming. We agree that a new NN model has to be trained for each optimization objective: each objective f has its own optimal preconditioner ∇f*.
>
>
> Q: The function of x and the gradient is complex, it is difficult to predict the relationship by using a simple network.
> A: Neural Networks might be our best shot at approaching ∇f* since they’re the most universal function approximators.

---

### Official Review · AnonReviewer1 · 2019-10-23
**Official Blind Review #1**

**Rating:** 1

**Review:**

This paper attempts to learn a preconditioner for optimization, specifically for the Dual space preconditioned descent (DPGD).
- The techniques used to learn the preconditioner are heuristic, not scalable and without justification or ablation studies.
- It does not compare against "standard" optimization techniques that construct data-driven preconditioners such as Adam or Adagrad or even to more Newton, natural gradient methods that use the Hessian or the Fisher information matrix as preconditioners. It shows ad-hoc synthetic experiments in dimensions 1 and 50. This is clearly not enough.
Detailed review below:
- Section 2: Please explain why Legendre functions are useful in ML. For assumption 1, 2; it needs to be explained why these hold for a given f*. What constraints do you need on f? What functions satisfy these? Please explain this explicitly.
- Section 3: What is the number of points x_i needed in high dimensions to learn? Is it even possible to scale up this method to high dimensions?
- Constructing \mu requires computing the determinant of the Jacobian. What is the computational complexity? Moreover, it seems that we need access to the \nabla f(x) for all x in D(f)?
- Please state all the assumptions in the beginning rather than introducing one at a time in the propositions.
- Remark 1: It is unclear that the cost of an inverse Hessian matrix is more than the procedure proposed in this paper.
- Section 3.5: Please explain what is the advantage of this learned optimizer compared to other methods? Note that there is literature on non-smooth optimization and methods like sub-gradient descent can be used in this case.
- What is the justification for the selection of the loss function and log-rescaling?
- The result of Lemma 1 is standard. Please acknowledge this.
-  Section 4: "The step-size is set to 1". It seems that the optimizer has been overfit and engineered to work on this specific problem. Either these decisions need to be justified, there needs to be an ablation study or there needs to be a larger set of experiments.




**Experience Assessment:**

I have published one or two papers in this area.

**Review Assessment: Checking Correctness Of Derivations And Theory:**

I carefully checked the derivations and theory.

**Review Assessment: Checking Correctness Of Experiments:**

I carefully checked the experiments.

**Review Assessment: Thoroughness In Paper Reading:**

I read the paper thoroughly.

---

> ### Author Response · Authors · 2019-11-14
> **Response to Review #1**
>
> Thank you for your careful analysis and feedback. We are adding more experiments and benchmarks in response to your comments.
>
> Q: The techniques used to learn the preconditioner are heuristic, not scalable and without justification or ablation studies.
> A: The specific algorithm we build upon (DPGD, Maddison et. al 2019) is currently difficult to use practice: there is no theoretical or heuristical way of selecting a pre-conditioner for this algorithm. We propose the first existing method to chose such a pre-conditioner automatically. This method is motivated by a theoretical observation ($\nabla f^*$ is a perfect pre-conditioner for the DPGD algorithm) and gives unmatched results on specific optimization problems.
>
> Q: It does not compare against “standard” optimization techniques that construct data-driven preconditioners such as Adam or Adagrad or even to more Newton, natural gradient methods that use the Hessian or the Fisher information matrix as preconditioners.
> A: We agree that we could have compared our method with advanced optimization technics. However, the methods listed above might not be appropriate. Adam and Adagrad are known to be efficient with stochastic objectives (some convergence results even require stochasticity, cf arxiv: 1810.02263) which is not the setting of DPGD. Newton and natural gradient require matrix inversion at each step and are therefore even less scalable, but they would indeed have made good benchmarks.
>
> Q: Section 2: Please explain why Legendre functions are useful in ML. What constraints do you need on f? What functions satisfy these?
> A: Our objective is to efficiently solve convex optimization problems. This might seem restricted in the field of machine learning but has numerous applications. Legendre is a large class of convex functions. The Legendre assumption is one of the assumptions needed to ensure convergence of the dual-space preconditioning algorithm. It is the assumption made by Maddison et. al, 2019. We needed to make this assumption to be able to restate their results.
>
> Q: For assumption 1, 2; it needs to be explained why these hold for a given $f^*$.
> A: $f$ is Legendre iif  $f^*$ is Legendre (cf. Proposition 1)
>
> Q: Section 3: What is the number of points $x_i$ needed in high dimensions to learn?
> A: We do not have theoretical results, but this may be difficult to obtain in the context of deep learning.
>
> Q: Is it even possible to scale up this method to high dimensions?
> A: The method we use to correct the input distribution doesn’t scale. However, we only need this method when $f$ is non-smooth / non-strongly convex. In other cases, our algorithm does scale.
>
> Q: Constructing $\mu$ requires computing the determinant of the Jacobian. What is the computational complexity?
> A: The complexity varies depending on the nature of the objective function. It comes mainly from the computation of the determinant and the computation or approximation of the Jacobian.
>
> Q: Moreover, it seems that we need access to the $\nabla f(x)$ for all x in $D(f)$?
> A: This is indeed required for gradient descent. Note that we are doing deterministic optimization and therefore suppose we have access to exact gradients for every x in the dataset.
>
> Q: It is unclear that the cost of an inverse Hessian matrix is more than the procedure proposed in this paper.
> A: Supposing we have free access to the Hessian, matrix inversion is more expensive than matrix multiplication. A step requiring matrix inversion is, therefore, less expensive than a step requiring the evaluation of a neural network.
>
> Q: Please explain what is the advantage of this learned optimizer compared to other methods? Note that there is literature on non-smooth optimization and methods like sub-gradient descent can be used in this case.
> A: The subgradient algorithm typically requires the sublinearity of subgradients. Subgradient cannot be applied in the setting we consider, for example, it would not converge for a power function objective like $f(x) = x^{50}$. In such a case, because f is differentiable, a subgradient step is simply a gradient descent step, which does not converge in this case.
>
> Q: What is the justification for the selection of the loss function and log-rescaling?
> A: The MSE loss function is standard for regression. The justification for the choice of log-rescaling is in Appendix D. In summary, log-rescaling seems empirically adapted to rescale the power function dataset as it reduces the scale of large gradients but not of gradients close to zero.
>
> Q: The result of Lemma 1 is standard. Please acknowledge this.
> A: It is indeed standard, we will provide a reference.
>
> Q: Section 4: “The step-size is set to 1". It seems that the optimizer has been overfit and engineered to work on this specific problem.
> A: This choice is justified by theory: in the theoretical observation mentioned above, the perfect preconditioner $\nabla f^*$ is used with step-size 1.

---

### Official Review · AnonReviewer4 · 2019-11-07
**Official Blind Review #4**

**Rating:** 3

**Review:**

This paper proposes an optimization method with the preconditioning in the framework of supervised learning. The ideal preconditioning is given by the Fenchel conjugate of the optimization function. This paper uses a supervised scenario to find the ideal preconditioning. The authors point out the importance of the sampling distribution then and propose a sampling scheme using the uniform distribution on the space of gradient descents. The samples are used to train neural networks that imitate the mapping of the Fenchel conjugate. The training network is incorporated into the Dual space Preconditioned Gradient Descent (DPGD). Some numerical experiments show the effectiveness of the proposed method comparing to the standard gradient descent method.

The authors proposed an interesting approach to the preconditioning in optimization problems. However, the paper is not well-written. In particular, the optimization algorithm is not clearly described. Numerical experiments with some toy problems are not very convincing to show the benefit of the proposed method. Though this paper may have some interesting ideas, more intensive analysis would be required.

other comments:
- The optimization algorithm is not explicitly described. Is the neural network trained as the batch learning? Is it possible to use the learning of the preconditioning in an online manner?
- It is not sure whether the ideal distribution \mu over the domain D(f) presented in Proposition 3 is computationally tractable.
- In Proposition 3: I think that the uniform property of the sampling does not directly mean the optimality in the sense of the learning accuracy. The authors need to investigate the more detailed relationship between the distribution mu and the prediction accuracy of the Fenchel conjugate.
- The proposed method requires the learning of neural networks, which will be computationally demanding. Please report the overall computational cost of the optimization algorithm in numerical experiments.


**Experience Assessment:**

I have read many papers in this area.

**Review Assessment: Checking Correctness Of Derivations And Theory:**

I assessed the sensibility of the derivations and theory.

**Review Assessment: Checking Correctness Of Experiments:**

I did not assess the experiments.

**Review Assessment: Thoroughness In Paper Reading:**

I made a quick assessment of this paper.

---

> ### Author Response · Authors · 2019-11-14
> **Response to Review #4**
>
> Thank you for your careful analysis and feedback. We are adding more experiments and benchmarks in response to your comments.
>
> Q: The optimization algorithm is not explicitly described.
> A: The optimization algorithm is DPGD (Maddisson et al. 2019).
>
> Q: Is the neural network trained as the batch learning? Is it possible to use the learning of the preconditioning in an online manner?
> A: The neural network is trained before starting the optimization. An online adaptation of our method is an interesting avenue to explore which we did not address in this article.
>
> Q: It is not sure whether the ideal distribution $\mu$ over the domain $D(f)$ presented in Proposition 3 is computationally tractable.
> A: The complexity varies depending on the nature of the objective function. It comes mainly from the computation of the determinant and the computation or approximation of the Jacobian.
>
> Q: In Proposition 3: I think that the uniform property of the sampling does not directly mean the optimality in the sense of the learning accuracy. The authors need to investigate the more detailed relationship between the distribution mu and the prediction accuracy of the Fenchel conjugate.
> A: The uniformization of the samples is motivated by Y. Bengio’s “Practical recommendations for gradient-based training of deep architectures” and is supposed to improve the quality of the regression (just as having a balanced dataset helps when learning a classifier). This recommendation, in particular, seems to be empirical and we might indeed need to investigate if it holds in our case.
>
> Q: The proposed method requires the learning of neural networks, which will be computationally demanding. Please report the overall computational cost of the optimization algorithm in numerical experiments.
> A: We have reported the number of epochs needed to train our neural network as well as the structure of the neural network. For the power function, for instance, we trained a neural network of size $256\times 128$ on a data-set of $1000$ samples during $100$ epochs.

---

### Decision · Program_Chairs · 2019-12-19

**Decision:**

Reject

**Comment:**

Thanks for the detailed replies to the reviewers, which significantly helped us understand your paper better.
However, after all, we decided not to accept your paper due to weak justification and limited experimental validation. Writing should also be improved significantly. We hope that the feedback from the reviewers help you improve your paper for potential future submission.